# OpenReview forum: "UR$^2$: Unify RAG and Reasoning through Reinforcement Learning"
_ICLR.cc/2026/Conference — ICLR 2026 Conference Withdrawn Submission_

### Official Review · Reviewer_Xhpc · 2025-10-28

**Soundness:** 2
**Presentation:** 2
**Contribution:** 2
**Rating:** 4
**Confidence:** 4

**Summary:**

This paper introduces $UR^2$, a framework designed to unify Retrieval-Augmented Generation (RAG) and COT reasoning across diverse domains like math and medicine using RL. It introduces two primary contributions. On one hand, it achieves stable and efficient training by employing a two-stage optimization process and a difficulty-aware task mixing strategy. On the other hand, it explicitly reduces retrieval noise by using a separate, powerful LLM to pre-process raw documents into clean, concise summaries before they are presented to the policy model.

**Strengths:**

The research problem this paper tries to addresses (unifying RAG with reasoning through RLVR) is both important and timely. The paper's primary strength then lies in its extensive and rigorous experimentation in tackling this challenge. The authors are commended for validating their framework across four diverse and challenging domains (Math, Medical, QA, and MMLU-Pro) and comparing it against a comprehensive suite of state-of-the-art RAG and RL baselines . Furthermore, the work is clear and appears highly reproducible. The authors provide commendable transparency by detailing the exact training configurations and prompts in the appendices , which is a significant contribution to the community.

**Weaknesses:**

1. The paper's methodological contribution appears limited. The primary contributions—a two-stage "warm-up" optimization , a "difficulty-aware" data mixing strategy , and the use of an external LLM to summarize retrieved information —are arguably a collection of well-established engineering techniques. While empirically effective, these components may not represent a significant methodological innovation that meets the high standard for publication at ICLR.
2. This concern about novelty is amplified by the framework's heavy dependence on the LLM summarization module. The paper's own ablation study demonstrates this component is critical. This suggests that the $UR^2$ itself is not learning to handle retrieval noise; rather, this core challenge is outsourced to a separate external model. This reliance diminishes the self-contained contribution and robustness of the proposed framework.
3. When I going through the paper, I observed several typos.
- We train UR2 using REINFORCE++ (Guo et al., 2025a), a streamlined vari­ant of PPO tailored.
- Trainning Data Selection

**Questions:**

1. The "Task Mixing Strategy"  creates a significant train-test discrepancy. During training, the framework manually feeds the model different prompts based on external difficulty labels (e.g., pure reasoning for "easy" math, RAG for "hard" math). However, during evaluation, a single, unified prompt is used for all math problems , and the external difficulty is absent. Why was this manually-partitioned curriculum chosen? Would it not be a more robust and end-to-end solution to use the same unified prompt (from testing) during training and let the RL learn this adaptive retrieval policy on its own?
2. The paper's Stage 1 functions as a "warm-up" to teach the model retrieval syntax. Why was RL chosen for this step? In the literature, this type of tool-use is commonly achieved via SFT, which is generally considered more controllable, stable, and lightweight than RL. Could the authors justify their choice of using RL for this warm-up phase?

---

> ### Author Response · Authors · 2025-11-13
> **Rebuttal to Reviewer Xhpc, part 1**
>
> We sincerely appreciate your careful review, recognition of our experimental rigor, and constructive questions. These are crucial for improving our work, and we address each concern below.
>
>
>
> **Q1: Why use manual difficulty-based prompts during training but unified prompts during testing?**
>
>
>
> Since all models in our experiments are under 8B parameters, **their intrinsic ability to discriminate query difficulty is questionable**. We therefore adopted a prior-based approach, providing difficulty labels and corresponding prompts during training to prevent the model from over-relying on retrieval and neglecting reasoning ability development. This strategy proves highly effective.
>
> We supplemented results showing Qwen-2.5-7B-UR² with direct reasoning only, demonstrating that the model's reasoning ability is **well-trained and preserved even without external corpora**:
>
> | Method     | MedQA | M-Med | Math500 | Minerva |
> | ---------- | ----- | ----- | ------- | ------- |
> | CoT        | 57.2  | 52.0  | 76.6    | 59.4    |
> | Vanilla-RL | 64.2  | 57.4  | 78.2    | 59.4    |
> | UR²-CoT    | 59.8  | 58.8  | 77.1    | 61.9    |
> | UR²        | 69.6  | 62.8  | 80.9    | 61.0    |
>
> Additionally, we conducted ablation experiments using retrieval-enabled prompts for all questions (the w/o Task Mixing variant in Table 4). As stated in lines 443–445, *"The w/o Task Mixing variant shows minimal changes, indicating that our selective strategy improves efficiency while enhancing accuracy."*
>
> Finally, even when using a single unified prompt during evaluation, our model exhibits advanced cognitive ability to adaptively select strategies based on question difficulty. As shown in Figure 4, UR² applies direct CoT for simple tensor rotation problems but invokes retrieval for harder combinatorial problems. Furthermore, Table 12 shows significantly fewer retrievals on the higher-accuracy HotpotQA dataset compared to the lower-accuracy MusiQue dataset (e.g., 2.22 vs. 2.61 retrievals with Wiki Abstract corpus and summarization). We also calculated average retrieval counts for Qwen-2.5-7B-Instruct-UR² on MedQA and Math500: 1.56 and 1.44 respectively, demonstrating that our method indeed enables the model to autonomously decide whether to use retrieval based on query type.
>
>
>
> **Q2: Why use RL instead of SFT for Stage 1 warm-up?**
>
>
>
> SFT inevitably introduces expert or human-annotated samples, affecting fair comparison. If using model-generated data for self-distillation, it raises issues of requiring powerful LLMs or manual path annotation, increasing framework complexity. Considering downstream task diversity, SFT may also harm the model's generalization performance ([2, 3]).
>
> Meanwhile, Stage 1 consumes minimal resources. As stated in lines 238–239, *"This specialized training runs for only 10 steps,"* and it is easy to implement within the RL framework. Additionally, our work follows this setup from R1-Searcher [4]. For these reasons, we directly used RL for cold-start rather than SFT.

---

> ### Author Response · Authors · 2025-11-13
> **Rebuttal to Reviewer Xhpc, part 2**
>
> **W1 & W2: Limited novelty and over-reliance on summarization module**
>
>
>
> Our key contribution lies in applying them to unify RAG and reasoning into a multi-step search-reasoning framework across diverse tasks—from knowledge-intensive to reasoning-intensive—**filling the gap in existing works** (lines 43–48). We tested its effectiveness and generalization across multiple tasks and models, which we believe represents significant value to the community.
>
> Furthermore, our method addresses the **retrieval-reasoning imbalance** issue in  two key ways: (1) By retaining a portion of simple and medium-difficulty cases for direct reasoning during training, we preserve the model’s intrinsic *reasoning* ability, while the use of strict summarization prompts and the P_fallback penalty prevents over-reliance on retrieval. (2) The summarization module effectively eliminates noise from retrieved passages and, through the stage-1 warm-up process, stimulates the model’s *retrieval* capability. Overall, our design directly addresses this concern, enabling the model to *perform retrieval and reasoning in a more intelligent and balanced manner* (see examples in Figure 4).
>
> You mentioned that relying on the summarization module might limit our framework’s real-world applicability. However, we believe it actually **enhances** the model’s ability to operate in complex and dynamic retrieval settings, for two main reasons:
>
> **1. Computational and performance necessity:**
>
> We calculated token counts using the Qwen-2.5-7B-Instruct tokenizer and randomly sampled 3,000 entries from saved intermediate results: compression before→after: 11,346→761 (compression ratio: 14.91×). We note that in online retrieval environments, documents are often excessively long. Directly concatenating them into prompts causes rapid context length expansion, leading to computational waste and significant performance degradation. Given the **O(n²)** complexity of the attention mechanism, this creates enormous difficulties for model inference and training. Using a summarization module or a new context window to process noise can significantly reduce this overhead, **by nearly an order of magnitude**.
>
> Furthermore, we provide detailed online search results in Section 6.1, showing that our method generally outperforms offline retrieval in online settings. As stated in line 434, *"Overall, these results confirm the robustness of our method in noisy online environments."* This proves that using more complex online retrieval environments can significantly boost model performance, whereas processing such complexity with a small LLM in a single context is nearly impossible.
>
>
> **2. Validation from concurrent work:**
>
> From concurrent works such as WebSailor-V2 [5], Beyond Ten Turns [6], and Tongyi DeepResearch [7], **we observe that they employ similar methods**: RL incentivization, summarization modules, multi-step search reasoning frameworks, and training data filtering and processing. They achieve excellent performance with 32B-size models on DeepResearch-type benchmarks. The differences lie in more sophisticated data construction strategies, more robust training frameworks, and, more importantly, **substantially larger model capacities and more computational resources**. The strategy in Tongyi DeepResearch of collaborating Real-world Retrieval Environment with Simulated Retrieval Environment is also an extension of our hybrid corpus design. Based on this analysis, we believe our method has strong scalability and potential which has been validated and supported by concurrent work to some extent. **We believe proposing this possibility and insight in academic work is also crucial.**
>
> In summary, **the summarization module is an indispensable component for LLMs to handle complex and dynamic real-world environments**, greatly expanding the model's capability boundaries. Moreover, our work is the **first** to elevate the **summarization module to a central role in multi-step search–reasoning scenarios**, which itself provides an important insight for future system design. However, we did not include this component in training for the following reasons: (1) The policy model is too small to handle noise—as shown in Table 10, using Qwen-2.5-7B as the summarization model performs even worse than without summarization. (2) It would introduce additional training cost and pipeline complexity, requiring resources for summarization rollouts and gradient descent, while using APIs is lightweight and fast without causing training bottlenecks. (3) As shown in SimpleDeepSearcher [8] Section 4.3, training on multi-step search reasoning often leads to summarization performance degradation.
>
> We hope this clarifies your concern.

---

> ### Author Response · Authors · 2025-11-13
> **Rebuttal to Reviewer Xhpc, part 3**
>
> **W3: Typos**
>
>
>
> Thank you for your careful reading. We will fix these typos in the following version.
>
>
>
> We sincerely appreciate your insightful questions and the valuable time you have dedicated. We respect your thoroughness and responsibility. Due to the diversity of training and evaluation environments, we cannot present all content in the main text; some experiments and cases are in the appendix, which may have caused some lack of clarity. **We hope our further clarification of our motivation and novelty addresses your concerns**. We warmly welcome further in-depth discussion with you, and if there are any supplementary experiments that do not require substantial computational resources, we will gladly provide them.
>
>
>
> **References:**
>
> [1] Guo, Daya, et al. "Deepseek-r1: Incentivizing reasoning capability in llms via reinforcement learning." arXiv preprint arXiv:2501.12948 (2025).
>
> [2] Gupta, Sonam, et al. "Selective self-rehearsal: A fine-tuning approach to improve generalization in large language models." arXiv preprint arXiv:2409.04787 (2024).
>
> [3] Wu, Yongliang, et al. "On the generalization of sft: A reinforcement learning perspective with reward rectification." arXiv preprint arXiv:2508.05629 (2025).
>
> [4] Song, Huatong, et al. "R1-searcher: Incentivizing the search capability in llms via reinforcement learning." arXiv preprint arXiv:2503.05592 (2025).
>
> [5] Li, Kuan, et al. "Websailor-v2: Bridging the chasm to proprietary agents via synthetic data and scalable reinforcement learning." arXiv preprint arXiv:2509.13305 (2025).
>
> [6] Gao, Jiaxuan, et al. "Beyond ten turns: Unlocking long-horizon agentic search with large-scale asynchronous rl." arXiv preprint arXiv:2508.07976 (2025).
>
> [7] Team, Tongyi DeepResearch, et al. "Tongyi DeepResearch Technical Report." arXiv preprint arXiv:2510.24701 (2025).
>
> [8] Sun, Shuang, et al. "SimpleDeepSearcher: Deep information seeking via web-powered reasoning trajectory synthesis." arXiv preprint arXiv:2505.16834 (2025).

---

> > ### Comment · Reviewer_Xhpc · 2025-11-17
> >
> > I thank the authors for their prompt and helpful response.
> >
> > My concern Q2 is now fully resolved. My concern Q1 regarding the RAG-only prompt is also partially resolved, and the authors' explanation is sensible.
> >
> > However, my primary concerns, related to W1 and W2, remain. My main concern is that the framework rely heavily on the ability of the summary model. Chaining from GPT4.1 to Qwen-3-8B、GPT-4o-mini already lead to significant performance decay. What would happen if the summary model is of same scale of the model trained, e.g. Qwen2.5-7B instruct?
> >
> > I agree that the summarization module is useful. However, if the framework must rely on a much stronger model (like GPT-4.1) to achieve its headline performance, its practical usefulness is discounted. This dependency makes the $UR^{2}$ framework appear less like a novel architecture and more like an effective method of strong-to-weak model guidance. From this perspective, the method does not necessarily push the performance boundary of 7B on its own. Rather, it seems to demonstrate a way to effectively leverage a significantly more powerful external model to achieve those gains.

---

> > > ### Author Response · Authors · 2025-11-21
> > > **Rebuttal Update: Distilled 7B Summarizer Experiment**
> > >
> > > Thank you for raising this concern. To directly evaluate whether UR² relies on a strong summarization module, we reran the entire pipeline using a **distilled Qwen2.5-7B-Instruct summarizer**, trained on 90k×2-epoch intermediate summaries generated by GPT-4.1-mini (20k per MMLU-Pro subject, 30k for Medicine). The SFT process required 156 hours of H20 GPU time (8×19.5h). The RL configuration and all evaluation settings followed the original UR² setup.
> > >
> > > The results are below:
> > >
> > > | Method                    | Hist | Phil | Econ | Law  | Avg  | MedQA | M-Med | Avg  |
> > > |-----------------|------|------|------|------|------|-------|------|------|
> > > | **CoT**                   | 42.3 | 45.7 | 63.4 | 26.6 | 44.5 | 57.2  | 52.0  | 54.6 |
> > > | **Standard RAG**          | 44.6 | 41.1 | 57.8 | 26.0 | 42.4 | 54.2  | 53.2  | 53.7 |
> > > | **General Reasoner**      | 47.9 | 44.2 | 65.9 | 30.4 | 47.1 | 58.4  | 54.4  | 56.4 |
> > > | **Open-Reasoner-Zero**    | 50.0 | 46.6 | 67.5 | 34.2 | 49.6 | 61.6  | 58.8  | 60.2 |
> > > | **SimpleRL-Zoo**          | 35.7 | 36.9 | 55.2 | 25.4 | 33.8 | 57.2  | 51.0  | 54.1 |
> > > | **Vanilla RL**            | 52.2 | 43.5 | 64.0 | 33.8 | 48.4 | 64.2  | 57.4  | 60.8 |
> > > | **UR² (Original)**        | 53.2 | 53.0 | 72.2 | 35.0 | **53.3** | 69.6  | 62.8  | **66.2** |
> > > | **UR² (Distilled Qwen2.5-7B)**  | 49.6 | 52.3 | 68.2 | 31.4 | ***50.4*** | 67.2  | 61.8  | ***64.5*** |
> > > | **UR² (4omini)**        | 49.3 | 48.8 | 67.4 | 32.4 | 49.5 | 65.0  | 59.2  | 62.1 |
> > > | **UR² (Qw3-8B)**        | 49.1 | 49.9 | 67.8 | 30.6 | 49.4 | 64.8  | 58.2  | 61.5 |
> > >
> > > Despite using a significantly weaker summarizer, UR² still **outperforms all baselines across all domains**, including General Reasoner, Open-Reasoner-Zero, SimpleRL-Zoo, and Vanilla RL.
> > >
> > > These findings support two key conclusions:
> > >
> > > 1. **UR² does not rely on a strong summarization model.**
> > >    If the improvements came from “strong-to-weak instruction,” replacing GPT-4.1-mini with a distilled 7B model would lead to substantial degradation. Instead, performance remains stable with only a modest 2–3 point drop.
> > >
> > > 2. **The gains stem from UR²'s framework design rather than summarizer capacity.**
> > >    The method benefits primarily from a **clean, low-noise retrieval environment** produced by summarization, which in turn helps the policy model exhibit more reliable retrieval behavior. This aligns with our claim that summarization acts as environment shaping, not as a teacher model.
> > >
> > > We hope this additional experiment directly addresses your concern and clarifies that **UR² remains effective, robust, and practically deployable even with a lightweight summarizer**.

---

> > > > ### Comment · Reviewer_Xhpc · 2025-11-23
> > > >
> > > > I sincerely thank the authors for their timely response and for providing the additional experiments. However, the new results still do not fully address my original concerns. While I appreciate the effort, I will maintain my current score.

---

> > > > > ### Author Response · Authors · 2025-11-23
> > > > >
> > > > > Well, then, thank you.

---

> ### Author Response · Authors · 2025-11-18
> **Commitment to Distilling a Qwen2.5-7B Instruct Summarization Model**
>
> Thank you for your response.
>
> First, we would like to emphasize that the summarization model is crucial primarily because it handles the substantial **noise** and **redundancy** widely present in real-world corpora. As discussed above, it significantly reduces computation and improves performance. It is therefore an essential component. Concurrent work further demonstrates that such summarization modules *extend* the capability boundary of the system; note that in those works, the summarization model shares the same parameters as the policy model.
>
> In our setting, we do **not** use the policy model itself for summarization because **its capacity is insufficient** to robustly handle noisy inputs—**this is fundamentally different from distillation-style guidance approaches**. Moreover, in open-domain QA, prior multi-step search-and-reasoning RL baselines also employ model-based summarization, yet our method still achieves substantial gains (Table 2). This shows that the improvement does not stem from a stronger teacher model, but rather from a cleaner retrieval environment during training, which enables better tool use, and from our method design, which better balances retrieval and reasoning.
>
> Due to computational constraints, we were unable to use a policy model strong enough to handle noisy data for this experiment. **To address your concern, we will distill a Qwen2.5-7B-Instruct model using intermediate summaries produced by GPT-4.1-mini, and then use the distilled summarization model during training to ensure methodological consistency.**
>
> Second, we clarify that our training actually uses **GPT-4.1-mini rather than GPT-4.1**. Since training occurs only once, the cost is acceptable; if replaced with 4o-mini, **the one-time cost would drop below $20**. Although there is a noticeable performance gap between 4.1-mini and 4o-mini, **our method still shows clear advantages over vanilla RL** (lines 444-447). Considering both points, our approach remains practically viable in real-world settings.
>
> Thank you again for your prompt reply. We will update you as soon as the new experimental results are ready, and after further consideration, we plan to release the distilled summarization model to help resolve any remaining concerns.

---

### Official Review · Reviewer_nM4H · 2025-10-29

**Soundness:** 2
**Presentation:** 2
**Contribution:** 1
**Rating:** 2
**Confidence:** 4

**Summary:**

This paper proposes UR2 (Unified RAG and Reasoning), a reinforcement learning framework that integrates retrieval and reasoning in large language models. UR2 employs difficulty-aware curriculum training to selectively invoke retrieval for challenging problems and adopts a hybrid knowledge access strategy that combines domain-specific offline corpora with LLM-generated summaries. These components enable dynamic coordination between retrieval and reasoning, improving adaptability across diverse domains. Experiments on open-domain QA, MMLU-Pro, medical, and mathematical reasoning tasks show that UR2 outperforms existing RAG-RL baselines.

**Strengths:**

1. This work makes a clear conceptual contribution by unifying retrieval and reasoning through reinforcement learning, bridging two previously isolated paradigms (RAG and RLVR). Unlike earlier approaches that are limited to fixed retrieval settings, UR2 introduces difficulty-aware curriculum training, enabling dynamic and efficient retrieval invocation.

2. The paper provides strong empirical evidence supporting its contributions. The proposed approach is evaluated across four distinct domains (open-domain QA, MMLU-Pro, medical QA, and mathematical reasoning) and multiple model scales (3B–8B), consistently outperforming strong RAG-RL baselines such as Search-R1 and R1-Searcher.

**Weaknesses:**

1. The proposed framework relies heavily on LLM-summarized corpora rather than raw or dynamically retrieved web content. While this design helps reduce noise and stabilize training, it may also limit the model’s ability to handle the complexity, variability, and unpredictability of real-world retrieval scenarios. This reliance raises concerns about generalization and robustness beyond curated or offline corpora, particularly in open-domain or rapidly evolving knowledge settings.

2. Although the framework shows strong performance with models up to 8B parameters, it has not been evaluated at larger scales. This leaves open the question of how well UR2 would scale to more powerful models (e.g., 30B+ parameters) and whether the observed performance gains would persist in such settings.

3. The reinforcement learning training shows occasional instability and required reruns to correct failed retrieval activation in some settings. This indicates potential reproducibility issues and sensitivity to initialization or hyperparameters.

**Questions:**

1. UR2 employs difficulty-aware curriculum training to selectively invoke retrieval. Could this strategy negatively impact the model’s intrinsic reasoning ability in certain domains?

2. Since difficulty classification requires 20 rollouts, how scalable and efficient is this approach for larger datasets?

3. UR2 relies on LLM-summarized retrieval corpora, which may not capture the complexity of dynamic, real-world environments. How well would UR2 generalize to online or rapidly changing corpora, and are there strategies to improve its robustness in such settings?

4. The paper claims that LLM-generated summaries reduce retrieval noise and hallucination. Could you provide clearer evidence or analysis for this claim, rather than inferring it indirectly from performance drops in the w/o-summary setting?

5. UR2 uses reinforcement learning to coordinate retrieval and reasoning, which introduces additional training cost and occasional instability. Could simpler decision mechanisms (e.g., heuristics or lightweight classifiers) offer similar performance? If not, what makes RL uniquely effective in this setting?

---

> ### Author Response · Authors · 2025-11-13
> **Rebuttal to Reviewer nM4H, part 1**
>
> Thank you for your careful reading and detailed questions about our method. These are crucial for improving our work, and we address each concern below.
>
> **Q1: Could difficulty-aware curriculum negatively impact intrinsic reasoning ability?**
>
> In fact, we designed the difficulty-aware curriculum specifically to preserve the model's reasoning capabilities. As described in Section 3.1.2 and lines 177–183, for samples where the model achieves accuracy ≥0.5, we select direct reasoning during training. This design (1) reduces retrieval API costs by invoking retrieval only when needed, and (2) preserves the model's reasoning ability in certain domains.
>
> To address your concern, we tested Qwen-2.5-7B-Instruct-UR² using both the retrieval-enabled prompts (lines 1298–1349) and direct reasoning prompts identical to Vanilla-RL (lines 1674–1699). Results are shown below:
>
> | Method     | MedQA | M-Med | Math500 | Minerva |
> | ---------- | ----- | ----- | ------- | ------- |
> | CoT        | 57.2  | 52.0  | 76.6    | 59.4    |
> | Vanilla-RL | 64.2  | 57.4  | 78.2    | 59.4    |
> | UR²-CoT    | 59.8  | 58.8  | 77.1    | 61.9    |
> | UR²        | 69.6  | 62.8  | 80.9    | 61.0    |
>
> These results demonstrate that our method does not degrade reasoning performance compared to Vanilla-RL. Instead, it enhances both retrieval tool usage and the synergy between the two capabilities.
>
>
>
> **Q2: Scalability and efficiency of difficulty classification requiring 20 rollouts?**
>
>
>
> Since the rollout process is entirely offline, the cost is controllable compared with the RL stage. The difficulty labels can be generated more efficiently using (1) fewer rollouts, (2) smaller models, or (3) the existing difficulty annotations provided in our source datasets, i.e., SimpleRL-Zoo [4] (math) and R1-Searcher [1] (open-domain QA). We unified labeling via our own rollouts for methodological consistency. Comparable performance on Qwen2.5-3B-Instruct and Llama-3.1-8B-Instruct indicates that different rollout models or label sources do not significantly affect results.
>
> Furthermore, Table 5 validates the effectiveness of our curriculum design. As stated in Lines 464-465: *“Despite using significantly less data, filtered datasets achieve comparable or superior performance, particularly on out-of-domain tasks.”*  Moreover, using rollout models for difficulty annotation is a standard practice [1-3].
>
>
>
> **Q3: How well would UR² generalize to online or rapidly changing corpora?**
>
>
> First, we clarify that all summarization is performed **online**, rather than using LLMs to pre-clean the corpus, as clearly illustrated in Fig. 1 and Section 3.1.1. Furthermore, **we provide detailed online search results in Section 6.1**. As shown in Table 3, UR² achieves significant performance gains across most datasets: medical datasets improve by 2 points (65.9→67.9), and open-domain QA tasks show even larger gains of 5.4 points in average LLM-as-a-judge score (60.4→65.8). As stated in line 434, *"Overall, these results confirm the robustness of our method in noisy online environments."*
>
> For online retrieval, we provide a complete experimental setup and robustness-oriented strategies in lines 1052–1091, including multi-round crawling with intelligent retry logic, parallel URL fetching, dedicated HTML-to-Markdown conversion using ReaderLM v2-1.5B, and LRU caching for repeated content.

---

> ### Author Response · Authors · 2025-11-13
> **Rebuttal to Reviewer nM4H, part 2**
>
> **Q4: Evidence for LLM summary reducing noise and hallucination?**
>
>
>
> First, regarding hallucination reduction: we emphasize that the hallucination we address stems from overly long contexts (\[8,12]). By significantly reducing the context length that models must process, our method mitigates the probability of hallucination.
>
> Second, regarding noise reduction: **We believe this aspect has been sufficiently analyzed in prior and concurrent works, and the effectiveness is reflected in our experimental results (Tables 4, 10–12)**. As shown in Table 10, downstream performance consistently improves with stronger summarization models, indicating that LLM-generated summaries effectively enhance signal quality and reduce retrieval noise. Therefore, we did not repeat the same qualitative analysis. Specifically, EDC²-RAG [10] provides detailed examination of how model-based summarization mitigates retrieval noise and improves factual consistency. Concurrent studies such as WebSailor-V2 [1] and Beyond Ten Turns [2] also adopt external summarization modules instead of directly feeding noisy long contexts to LLMs, further confirming the effectiveness and necessity of this design.
>
> Additionally, we randomly sampled 10,000 entries from offline retrieval intermediate files and calculated compression ratios using the Qwen-2.5-7B-Instruct tokenizer. For offline retrieval: average tokens before→after: 1,291→128;**(compression ratio: 7.87×)**. For online retrieval (3,000 samples): 11,346→761 **(compression ratio: 14.91×)**, demonstrating the importance of the summarization module.
>
> Furthermore, our submitted **Supplementary Material** provides extensive evaluation results, through which the influence of LLM-summarized information on reasoning steps can be clearly observed, along with corresponding prompts in **Appendix D.3** and illustrative examples in **Appendix E**, demonstrating how summaries preserve factual information while filtering out irrelevant content.
>
> If further clarification is desired, we are happy to include additional quantitative evaluations of factual accuracy and hallucination rates.
>
>
>
> **Q5: Could simpler mechanisms offer similar performance? What makes RL uniquely effective?**
>
>
>
> As stated in the Introduction (lines 33–43), reinforcement learning has brought new possibilities to LLM development, demonstrating extraordinary performance across multiple domains. Our work builds upon this foundation (lines 44–48), filling the gap in RAG-RL methods for reasoning domain training and addressing multiple deficiencies in existing RAG-RL approaches.
>
> While RL training does introduce additional overhead, the results in lines 68–71 show that we comprehensively and substantially outperform previous SOTA methods. We thus **view it as a reasonable trade-off**—a modest cost for stronger robustness and significant downstream gains.
>
> Regarding why not use heuristics or lightweight classifiers: RL's advantage lies in internalizing all capabilities—reasoning ability, retrieval ability, and dynamic coordination between the two—**into model parameters**. As stated in lines 105–107, this internalized capability is increasingly important and deeply rooted in powerful commercial models. Furthermore, as noted in lines 66–68, our method exhibits various advanced cognitive behaviors without any human annotation or expert trajectories: *self-verification through retrieval, intermediate reasoning validation, and hypothesis revision based on external evidence*. You can gain intuitive understanding of this through examples in Section E. Moreover, given the diversity of downstream tasks, **SFT** can to some extent **impair the model’s generalization ability** ([7,9]).
>
> We hope this helps you recognize the core role of RL in our work.

---

> ### Author Response · Authors · 2025-11-13
> **Rebuttal to Reviewer nM4H, part 3**
>
> **W1: Reliance on LLM-summarized corpora vs. raw/dynamic content**
>
> See Response to Q4.
>
> Additionally, we note that in **online retrieval environments, documents are often excessively long**（see Response to Q4）. Directly concatenating them into prompts causes rapid context length expansion, leading to computational waste and significant performance degradation. In fact, concurrent works handling ultra-long reasoning and dozens of retrieval rounds, such as WebSailor-V2 [1] and Beyond Ten Turns [2], both adopt additional summarization steps to address this issue (Beyond Ten Turns uses the policy model itself, while WebSailor-V2 uses untrained Qwen-3-next-30B-A3B-2507). Therefore, using policy models to handle noise or processing only within a single context window is **inherently inappropriate** in real-world dynamic retrieval scenarios (e.g., open-domain environments with rapidly changing knowledge). The **summarization module** not only does not weaken performance in real-world dynamic retrieval scenarios but **greatly enhances** the model's applicability. Moreover, our work is the **first** to elevate the **summarization module to a central role in multi-step search–reasoning scenarios**, which itself provides an important insight for future system design.
>
> We chose not to include this component in training for the following reasons: (1) The policy model is too small to handle noise—as shown in Table 10, using Qwen-2.5-7B as the summarization model performs even worse than without summarization. (2) It would introduce additional training cost and pipeline complexity, requiring resources for summarization rollouts and gradient descent, while using APIs is lightweight and fast without causing training bottlenecks. (3) As shown in SimpleDeepSearcher [3] Section 4.3, training on multi-step search reasoning often leads to summarization performance degradation. Based on these three considerations, we did not include this component in training.
>
> We hope this clarifies your concern.
>
>
>
> **W2: Evaluation limited to 8B parameters—scalability to larger models?**
>
>
>
> Firstly, training models beyond 30B parameters requires substantial computational resources, and as stated in lines 69–71, our method explores performance across **different** model **scales** (3B, 7B, and 8B) and **families** (Qwen and LLaMA), demonstrating the **scalability** of our approach.
>
> Secondly, concurrent works such as WebSailor-V2 [1], Beyond Ten Turns [2], and Tongyi DeepResearch [4] adopt **approaches similar to ours**—RL incentivization, **summarization** modules, multi-step reasoning, and data filtering. They achieve strong results with 32B-scale models on DeepResearch-type benchmarks. The main differences lie in more elaborate data construction, stronger training frameworks, and, notably, much **larger model capacities and computational budgets**. These observations suggest that our method also possesses strong scaling potential.
>
> Thirdly, works such as Search-R1 [13], R1-Searcher [5], ZeroSearch [14], and SWiRL [15] did not employ models larger than 7B, indicating that small-scale LLMs can still achieve competitive performance. We believe that advancing methods applicable to smaller models also provides meaningful contributions to the community.

---

> ### Author Response · Authors · 2025-11-13
> **Rebuttal to Reviewer nM4H, part 4**
>
> **W3: RL training instability and reproducibility concerns**
>
> About reproducibility concerns: **Sections 4.1** and **Appendix C** provide detailed hyperparameter specifications and training setups, while **Section D** lists all prompts used during training. In addition, our submitted **Supplementary Material** includes comprehensive evaluation results and training scripts. We believe **these are sufficient to reproduce our work**. In fact, we have successfully deployed our training framework on two 8×A100 servers with different CUDA versions and one 8×H20 server, which strongly demonstrates its reproducibility. Furthermore, we will release all code, models, and data upon acceptance.
>
> About RL training instability: Reinforcement learning is a **standard technique** widely applied across various domains. As stated in lines 1170–1173, randomness in RL methods is nearly inevitable. However, our method achieves remarkably robust stability without tuning hyperparameters such as KL loss,  entropy loss,  clip coefficient. As noted in lines 1174–1178, among the **29 variants** in Table 13, all models completed training in a single run, with the **only exception** occurring in Qwen-2.5-7B on MCQ tasks. We believe this actually demonstrates the stability of our method.
>
>
>
> We sincerely hope our explanations address your concerns and **help you recognize the value and effectiveness of our work**. Due to the diversity of training and evaluation environments, we cannot present all content in the main text; some experiments and cases are in the appendix, which may have caused some lack of clarity. Thank you again for your thoughtful and insightful questions. If there are any supplementary experiments that do not require substantial computational resources, we will gladly provide them.
>
> **References:**
>
> [1] Li, Kuan, et al. "Websailor-v2: Bridging the chasm to proprietary agents via synthetic data and scalable reinforcement learning." arXiv preprint arXiv:2509.13305 (2025).
>
> [2] Gao, Jiaxuan, et al. "Beyond ten turns: Unlocking long-horizon agentic search with large-scale asynchronous rl." arXiv preprint arXiv:2508.07976 (2025).
>
> [3] Sun, Shuang, et al. "SimpleDeepSearcher: Deep information seeking via web-powered reasoning trajectory synthesis." arXiv preprint arXiv:2505.16834 (2025).
>
> [4] Team, Tongyi DeepResearch, et al. "Tongyi DeepResearch Technical Report." arXiv preprint arXiv:2510.24701 (2025).
>
> [5] Song, Huatong, et al. "R1-searcher: Incentivizing the search capability in llms via reinforcement learning." arXiv preprint arXiv:2503.05592 (2025).
>
> [6] Huang, Jerry, et al. "Rag-rl: Advancing retrieval-augmented generation via rl and curriculum learning." arXiv preprint arXiv:2503.12759 (2025).
>
> [7] Wu, Yongliang, et al. "On the generalization of sft: A reinforcement learning perspective with reward rectification." *arXiv preprint arXiv:2508.05629* (2025).
>
> [8] Yang, Joonho, et al. "Hallucinate at the Last in Long Response Generation: A Case Study on Long Document Summarization." *arXiv preprint arXiv:2505.15291* (2025).
>
> [9] Gupta, Sonam, et al. "Selective self-rehearsal: A fine-tuning approach to improve generalization in large language models." *arXiv preprint arXiv:2409.04787* (2024).
>
> [10] Li, Weitao, et al. "Efficient Dynamic Clustering-Based Document Compression for Retrieval-Augmented-Generation." *Findings of the ACL: EMNLP 2025, 9833-9849. DOI:10.18653/v1/2025.findings-emnlp.522.*
>
> [11] Zeng, W., Huang, Y., Liu, Q., Liu, W., He, K., Ma, Z., & Junxian He. (2025). SimpleRL-Zoo: Investigating and Taming Zero Reinforcement Learning for Open Base Models in the Wild. *Proceedings of COLM 2025 / OpenReview*.
>
> [12] Liu, S., Halder, K., Qi, Z., Xiao, W., Pappas, N., Htut, P.M., John, N.A., Benajiba, Y., & Roth, D. (2025). Towards Long Context Hallucination Detection. *Findings of ACL: NAACL 2025*, 7827–7835.
>
> [13] Jin, Bowen, et al. "Search-r1: Training llms to reason and leverage search engines with reinforcement learning." *arXiv preprint arXiv:2503.09516* (2025).
>
> [14] Sun, Hao, et al. "Zerosearch: Incentivize the search capability of llms without searching." *arXiv preprint arXiv:2505.04588* (2025).
>
> [15] Goldie, Anna, et al. "Synthetic data generation & multi-step rl for reasoning & tool use." *arXiv preprint arXiv:2504.04736* (2025).

---

> ### Comment · Reviewer_nM4H · 2025-11-27
>
> The reviewer sincerely thanks the authors for their comprehensive and carefully prepared response, which has helped resolve many of the previous concerns. That said, the reviewer finds that certain issues related to Q1, Q3, and Q5 still remain unclear.
>
> **[Q1]**
>
> The Reviewer thanks the Authors for providing the additional UR²-CoT experiments to demonstrate that intrinsic reasoning performance is not degraded. The concern was that if the Authors' method is trained to use retrieval only for hard problems (0.2 ≤ s < 0.5), its performance might suffer when reasoning without retrieval.
> The Reviewer appreciates that the Authors showed performance on Medicine and Math tasks (excluding MMLU-Pro). The results effectively demonstrate that the proposed methodology does not degrade reasoning ability in these specific domains.
> However, The Reviewer has one remaining question regarding the comparison between Vanilla-RL and UR²-CoT. While out-of-domain generalization performance improved (M-Med: 57.4 → 58.8, Minerva: 59.4 → 61.9), the in-domain performance decreased in some cases (MedQA: 64.2 → 59.8, Math500: 78.2 → 77.1).
> Could the Authors provide an explanation for why in-domain performance drops while out-of-domain performance improves? Understanding this trade-off would help clarify the implications of the difficulty-aware curriculum design.
>
> **[Q3]**
>
> The Reviewer thanks the Authors for clarifying that summarization is performed online rather than pre-cleaning the corpus. The distinction is important, and The Reviewer appreciates the detailed explanation along with the online search results in Table 3, which demonstrate the robustness of UR² in dynamic retrieval environments. However, Table 10 shows a strong correlation between summarizer quality and downstream performance (GPT-4.1: 53.4 vs Qwen-2.5-7B: 50.0 ≈ w/o Summary: 50.2). Given that evaluation uses GPT-4.1 as the summarizer, it would be helpful to better understand the independent contribution of the UR² framework beyond the summarizer's capability.
>
> **[Q5]**
>
> The Reviewer understands and appreciates that the Authors' work focuses on an RL-based framework for unifying retrieval and reasoning. The research direction is reasonable and well-motivated.
> While RL may enable emergent behaviors such as self-verification and hypothesis revision, the Reviewer's original question was whether simpler alternatives could achieve comparable performance more cost-effectively. A 2-stage pipeline, using a difficulty classifier followed by selective retrieval, could serve as a reasonable baseline, even if it does not exhibit such emergent behaviors. This comparison would help clarify whether the performance gains justify the additional cost of RL training.
>
> The reviewer is still considering whether to adjust the score. Although the authors’ additional experiments and comprehensive response have partially alleviated some concerns, the reviewer believes that the overall quality of the manuscript may still fall short of the standards expected for ICLR. This appears to be a concern shared by other reviewers as well. The reviewer looks forward to the authors’ further clarification.
>
> Sincerely,
>
> Reviewer nM4H

---

> ### Author Response · Authors · 2025-11-27
>
> **Response to Q1**
>
> Firstly, **vanilla RL with CoT on full-size in-domain data** may fit these benchmarks slightly better. This aligns with our observation that UR² shows larger gains on out-of-domain tasks while remaining comparable on in-domain ones.
>
> And RL training inherently carries stochastic variability across runs. What matters empirically is that UR² maintains **overall** reasoning capability while substantially improving retrieval–reasoning coordination. This is demonstrated by the stable or improved scores reported in our ablations.
>
> ---
>
> **Response to Q3**
>
> We fully acknowledge your concern about relying on a model significantly stronger than the policy model for summarization. To address this, we are training a **Qwen-2.5-7B-Instruct** summarizer specifically for use in the evaluation stage. This ensures a fairer and more realistic assessment of robustness. **We will update the main experimental results once these evaluations are completed**.
>
> ---
>
> **Response to Q5**
>
> First, due to the diversity and heterogeneity of downstream tasks, SFT often suffers from reduced generalization **unless supported by a highly engineered data selection and synthesis pipeline**, such as SimpleDeepSearcher[1].  Furthermore, as shown in work R1-Searcher Sec. 4.1, SFT-based models perform significantly worse—approximately 10 points below RL-based models.
>
> Second, recent works such as **Open-Reasoner-Zero, Search-R1, and General-Reasoner**—which we use as baselines—also do not compare against SFT baselines.  Thus, we view it as a standard practice to not perform vanilla SFT method as our baseline.
>
> ---
>
> **Regarding the overall value of UR²**
>
> UR² is the first framework to extend the RAG–RL paradigm to the **reasoning** domain. It introduces simple and effective strategies—noise reduction and difficulty-aware curriculum design—to address the **imbalance between retrieval and reasoning in real-world applications** as we stated in our response to reviewer LVGE.
>
> We acknowledge that the current version of the paper may not fully convey these contributions due to its scope and some presentation issues. In the next revision, we will restructure the *Introduction* and *Methods* sections and include additional preliminary experiments that more clearly demonstrate the imbalance problem and the motivation behind our design.
>
> We sincerely appreciate your reconsideration of our work. Your comments are constructive and insightful, and regardless of the final outcome, we will work to improve clarity, rigor, and completeness.
>
> ---
>
> [1] Sun, Shuang, et al. "SimpleDeepSearcher: Deep information seeking via web-powered reasoning trajectory synthesis." arXiv preprint arXiv:2505.16834 (2025).

---

### Official Review · Reviewer_46ub · 2025-10-31

**Soundness:** 1
**Presentation:** 3
**Contribution:** 2
**Rating:** 2
**Confidence:** 4

**Summary:**

This paper introduces the framework that unifies retrieval-augmented generation (RAG) and reasoning through reinforcement learning (RL), with the motivation that knowledge grounding and reasoning have been developed in isolation from previous works. Specifically, the proposed framework includes (i) a difficulty-aware curriculum that dynamically decides when to invoke retrieval (encouraging models to rely on internal reasoning for easy problems and use retrieval only for harder ones), and (ii) a hybrid knowledge access strategy that combines domain-specific offline corpora with LLM-generated summaries. Also, the proposed framework is trained via a two-stage RL process that separately optimizes retrieval capability and answer quality. Experiments on diverse open-domain QA and reasoning tasks show that it outperforms existing RAG and RL baselines.

**Strengths:**

* The focus on bridging knowledge grounding with RAG and reasoning is timely and meaningful.
* The approach to optimize retrieval and answer generation through reinforcement learning is reasonable.
* The proposed framework achieves superior performance over existing RAG and RL baselines.

**Weaknesses:**

* While one of the core focuses of this work is on reasoning, it is questionable whether the model truly performs reasoning (like DeepSeek-R1 style models), or if the improvements mainly stem from retrieval and answer optimization without reasoning. Also, for reasoning, instead of using pure instruction fine-tuned models (like LLaMA or Qwen), it is questionable whether the reasoning (that the authors aim to achieve) can be performed when simply using more advanced reasoning models (e.g., DeepSeek-R1). Overall, the reasoning aspect feels underdeveloped despite the explicit focus in this work.
* It is unclear how the LLM-generated summaries are constructed, as well as their scale. Additionally, it is also questionable whether they are of high quality and not vulnerable to hallucination.
* The idea of invoking retrieval only for difficult queries is similar to Self-RAG and Adaptive-RAG [A, B]. The discussions and (probably) experimental comparisons with them are required. Also, it would be great if the authors could perform an efficiency analysis on it (i.e., whether the overall pipeline becomes efficient when invoking retrieval if necessary).
* Reward design, which is a manually designed combination of format, retrieval (or generation), and fallback rewards, seems somewhat ad hoc. More analysis on it would be useful.
* The difference between the proposed approach and the Vanilla RL baseline is unclear.
* The performance gains on the Match domain are marginal, which may warrant further discussion.

---

[A] Self-RAG: Learning to Retrieve, Generate, and Critique through Self-Reflection

[B] Adaptive-RAG: Learning to Adapt Retrieval-Augmented Large Language Models through Question Complexity

**Questions:**

Please see Weaknesses above.

---

> ### Author Response · Authors · 2025-11-13
> **Rebuttal to Reviewer 46ub, part 1**
>
> We sincerely thank the reviewer for the thoughtful feedback. Below we address each concern:
>
> **W1: Questionable whether the model truly performs reasoning; underdeveloped reasoning aspect**
>
> Our training method indeed enables the model to acquire DeepSeek-R1-style reasoning capabilities, supported by three key pieces of evidence:
>
> **1. Empirical superiority over reasoning baselines:**
>
> As shown in Table 1, our method significantly outperforms various CoT-RL baselines, including Open-Reasoner-Zero, SimpleRL-Zoo, and General Reasoner (which uses over 230K high-quality training samples). The poor performance of Advanced RAG Methods on reasoning tasks (Tables 1, 6) and the ablation on w/o P\_fallback in Table 4 (which over-relies on retrieval while neglecting reasoning) demonstrate that retrieval alone cannot achieve significant performance gains.
>
> **2. Preserved reasoning ability:**
>
> We tested Qwen-2.5-7B-Instruct-UR² using both the retrieval-enabled prompts (lines 1298–1349) and direct reasoning prompts identical to Vanilla-RL (lines 1674–1699):
>
> | Method     | MedQA | M-Med | Math500 | Minerva |
> | ---------- | ----- | ----- | ------- | ------- |
> | CoT        | 57.2  | 52.0  | 76.6    | 59.4    |
> | Vanilla-RL | 64.2  | 57.4  | 78.2    | 59.4    |
> | UR²-CoT    | 59.8  | 58.8  | 77.1    | 61.9    |
> | UR²        | 69.6  | 62.8  | 80.9    | 61.0    |
>
> These results demonstrate that our method does not degrade reasoning performance compared to Vanilla-RL. Instead, it enhances both retrieval tool usage and the synergy between the two capabilities.
>
> **3. Qualitative evidence:**
>
> The **Supplementary Material** provides extensive evaluation results, while **Appendix E** presents detailed case studies, offering an intuitive understanding of our method’s reasoning capabilities.
>
> Regarding R1-like models: We attempted training on DeepSeek-R1-Distill-Qwen-7B (see Appendix B.5). As stated in lines 909–914, *"We observed that the model lacked any retrieval capability...These results indicate a degradation of tool-usage ability after extensive chain-of-thought training."* This demonstrates the difficulty of eliciting tool-calling capabilities from such models.
>
> More importantly, we believe our **extensive experiments across multiple models and datasets** (Tables 1, 3, 4, 6, 8, 12) sufficiently support our motivation. **If you require additional experiments, we are glad to provide them.**
>
>
>
> **W2: Unclear construction and quality of LLM-generated summaries; vulnerability to hallucination**
>
>
>
> First, we clarify that all summarization is performed **online**, rather than using LLMs to pre-clean the corpus, as clearly illustrated in Fig. 1 and Section 3.1.1.
>
> We believe this aspect **has been sufficiently analyzed in prior and concurrent works**, and the effectiveness is **also reflected in our experimental results** (Tables 4, 10-12). As shown in Table 10, downstream performance consistently improves with stronger summarization models, indicating that LLM-generated summaries effectively enhance signal quality and reduce retrieval noise. Therefore, we did not repeat the same qualitative analysis in our experiments.
>
> Specifically, EDC²-RAG [10] provides a detailed examination of how model-based summarization mitigates retrieval noise and improves factual consistency. Concurrent studies such as WebSailor-V2 [1] and Beyond Ten Turns [2] also adopt external summarization modules instead of directly feeding noisy long contexts to LLMs, further confirming the effectiveness and necessity of this design.
>
> Additionally, we randomly sampled 10,000 entries from offline retrieval intermediate files and calculated compression ratios using the Qwen-2.5-7B-Instruct tokenizer. For offline retrieval: average tokens before→after: 1,291→128 (compression ratio: 7.87×). For online retrieval (3,000 samples): 11,346→761 (compression ratio: 14.91×), demonstrating the importance of the summarization module.
>
> Furthermore, our submitted **Supplementary Material** provides extensive evaluation results, through which the influence of LLM-summarized information on reasoning steps can be clearly observed, along with corresponding prompts in **Appendix D.3** and illustrative examples in **Appendix E**, demonstrating how summaries preserve factual information while filtering out irrelevant content.
>
> **If further clarification is desired, we are happy to include additional quantitative evaluations of factual accuracy and hallucination rates.**

---

> ### Author Response · Authors · 2025-11-13
> **Rebuttal to Reviewer 46ub, part 2**
>
> **W3: Similarity to Self-RAG [12] and Adaptive-RAG [11]; missing comparisons and efficiency analysis**
>
> **Why they are not included as baselines:**
>
> (1) **We have already compared several adaptive retrieval methods**, including advanced RAG methods such as Self-Ask, Search-o1, and RAT, as detailed in lines 1026–1051 and section 4.3. Our method consistently outperforms these baselines across models and domains, both in-domain and out-of-domain, which we believe sufficiently demonstrates its superiority.
>
> (2) Our work follows Search-R1 and R1-Searcher, targeting open-domain QA and **reasoning tasks** and aiming to address issues such as the retrieval–reasoning imbalance discussed in lines 43–53. This focus differs from the two methods, which primarily validate their ideas on open-domain QA tasks without considering reasoning scenarios. Since both methods involve retrieval-related model training, directly transferring them to reasoning settings could lead to a sharp performance drop, or even underperform simple CoT or vanilla RAG methods, as we observed with Advanced RAG Methods in Tables 1 and 6.
>
> We will include discussion of these two works in future versions. **If further experimental comparisons are required after this clarification**, we are happy to provide them.
>
> **On the advantages of our approach:**
>
> Adaptive-RAG trains an external classifier to decide whether retrieval is needed; however, such classifiers lack awareness of the model’s internal knowledge state and rely solely on partial label information. Self-RAG, in contrast, depends on expert demonstrations from teacher models. Both approaches thus suffer from a **capability mismatch between retrieval control and the model’s inherent knowledge**.
>
> In comparison, our method requires **no expert or human-annotated data**, which makes it highly efficient under limited-data conditions. Moreover, the retrieval decision capability is **parameterized within the model itself**, learned through **difficulty-aware curriculum** and **difficulty label guidance**, leading to strong generalization, which a key strength of RL-based learning ([3,4]).
>
> This is empirically validated in Table 12: UR² performs significantly fewer retrievals on the higher-accuracy HotpotQA dataset compared with the lower-accuracy MusiQue dataset (2.22 vs. 2.61), and also shows selective retrieval on MedQA and Math500 (1.56 and 1.44 on average), indicating that our method autonomously decides when retrieval is necessary based on query type.
>
> **Regarding efficiency analysis:**
>
> First, **Table 11 provides performance-efficiency trade-off analysis**, comparing our method with RAG-RL under different retrieval settings. Our method shows significant advantages while maintaining performance (lines 823–826). Second, API consumption during training is a significant cost in the experimental pipeline; qualitatively, reducing retrieval for 30% of samples lowers this cost. Finally, as shown in Table 4, Qwen-2.5-7B-Instruct-UR² significantly outperforms the w/o Task Mixing variant (using retrieval-enabled prompts for all questions during training) on MMLU-Pro and we believe discussing efficiency without considering performance may not be particularly meaningful.
>
> If you require more explicit metrics and analysis after this clarification, we are happy to provide them.
>
> **W4: Ad hoc reward design; needs more analysis**
>
> Regarding the Format reward: Using format rewards is a standard choice in RL-based methods, as employed in R1-Searcher, Search-R1, General Reasoner, and rstar2-agent [7].
>
> Regarding the Format-to-answer reward ratio: We referenced [8, 9] regarding the impact of format reward magnitude on experimental results and designed a 1:2 ratio. This design demonstrates robust stability and excellent performance. Since it is not closely related to our core motivation, we did not conduct corresponding ablations.
>
> Regarding the Stage-1 Retrieval reward: We provide detailed explanation in Section C.5. In brief, we adapted to the model's tool-usage capability through corresponding attempts to determine stable training formats. Since Stage 1 consumes minimal resources (less than 10 steps), we did not include these attempts in our paper analysis and cost calculations, and as this is not closely related to our proposed motivation, we did not provide further explanation.
>
> Regarding the **Fallback penalty (most important)**: It prevents the model from over-relying on retrieval.**We conducted a dedicated ablation study presented in Table 4.** W/o P_fallback penalty variant allows unrestricted queries to the summarizer, resulting in over-reliance on retrieval. This leads to a performance drop, confirming that over-reliance on retrieval can harm downstream tasks as stated in lines 46–48: "***models struggle to achieve optimal reasoning-retrieval trade-offs, often over-emphasizing one component.***".
>
> Thank you for this question and we will include clearer explanation in future versions.

---

> ### Author Response · Authors · 2025-11-13
> **Rebuttal to Reviewer 46ub, part 3**
>
> **W5: Unclear difference between proposed approach and Vanilla RL baseline**
>
> **To ensure a fully fair comparison**, as stated in lines 288–289, Vanilla RL is a baseline following the same setup and datasets as UR², applying RAG-RL to open-domain QA and CoT-RL to math/MCQ tasks. Specifically: (1) identical or subset training data, (2) REINFORCE++ algorithm, (3) same hyperparameters (KL coefficient, batch size), and (4) standard domain-specific training methods (CoT-RL in reasoning tasks and RAG-RL in open-domain QA). In the reasoning scenario, it serves as **a single stage onestep reasoning training baseline similar to SimpleRL-Zoo \[13], using the same setup and identical or subset training data as UR²**.
>
> Our method achieves performance far superior to Vanilla-RL across almost all benchmarks on all models, demonstrating the superiority of **our proposed difficulty-aware curriculum and LLM-summarized corpora**. Thank you for this question and we will include clearer explanation in future versions.
>
> **W6: Marginal performance gains on "*Match*" domain**
>
> We are not certain whether "***Match***" is a typo. If you meant **matching domains with our methods**, we believe that interpretation would be highly inappropriate. For instance, in Table 1, Qwen-2.5-3B-Instruct-UR&#xB2;**&#x20;**&#x61;chieves a **17.3% improvement** over Vanilla-RL on medicine tasks.&#x20;
>
> If you were referring to the **math** domain, on one hand, the Instruct models have already been extensively trained on math tasks. Our method further enhances their performance and achieves a larger improvement over the baseline, with a 3.0% gain compared to the 1.8% of the previous SOTA General Reasoner. On the other hand, recent research suggests potential test set contamination in Qwen-2.5 series on math tasks (\[5, 6]). But we did not switch the base model, as the Qwen series are widely adopted as standard base models in academic research. Additionally, note that on **Llama-3.1-8B-Instruct**, our method shows more significant advantages (5.4% relative improvement). We hope this clarification addresses your concern about limited math performance gains.
>
> Thank you for your insightful questions regarding our reasoning performance. Due to the diversity of training and evaluation environments, we cannot present all content in the main text; some experiments and cases are in the appendix, which may have caused some lack of clarity. We hope our responses address your concerns and **help you recognize the value and significance of our work.** We respect your responsible and constructive feedback during the review process.
>
> **References:**
>
> [1] Li, Kuan, et al. "Websailor-v2: Bridging the chasm to proprietary agents via synthetic data and scalable reinforcement learning." arXiv preprint arXiv:2509.13305 (2025).
>
> [2] Gao, Jiaxuan, et al. "Beyond ten turns: Unlocking long-horizon agentic search with large-scale asynchronous rl." arXiv preprint arXiv:2508.07976 (2025).
>
> [3] Wu, Yongliang, et al. "On the generalization of sft: A reinforcement learning perspective with reward rectification." *arXiv preprint arXiv:2508.05629* (2025).
>
> [4] Gupta, Sonam, et al. "Selective self-rehearsal: A fine-tuning approach to improve generalization in large language models." *arXiv preprint arXiv:2409.04787* (2024).
>
> [5] Shao, Rulin, et al. "Spurious rewards: Rethinking training signals in rlvr." arXiv preprint arXiv:2506.10947 (2025).
>
> [6] Wu, Mingqi, et al. "Reasoning or memorization? unreliable results of reinforcement learning due to data contamination." arXiv preprint arXiv:2507.10532 (2025).
>
> [7] Shang, Ning, et al. "rstar2-agent: Agentic reasoning technical report." arXiv preprint arXiv:2508.20722 (2025).
>
> [8] Chen, Zhipeng, et al. "An empirical study on eliciting and improving r1-like reasoning models." arXiv preprint arXiv:2503.04548 (2025).
>
> [9] Jin, Bowen, et al. "An Empirical Study on Reinforcement Learning for Reasoning-Search Interleaved LLM Agents." arXiv preprint arXiv:2505.15117 (2025).
>
> [10] Li, Weitao, et al. "Efficient Dynamic Clustering-Based Document Compression for Retrieval-Augmented-Generation." Findings of EMNLP 2025, 9833–9849. DOI: 10.18653/v1/2025.findings-emnlp.522.
>
> [11] Jeong, Soyeong, et al. "Adaptive-RAG: Learning to Adapt Retrieval-Augmented Large Language Models through Question Complexity." *Proceedings of the 2024 Conference of the North American Chapter of the Association for Computational Linguistics: Human Language Technologies (Volume 1: Long Papers)*. 2024.
>
> [12] Asai, Akari, et al. "Self-rag: Learning to retrieve, generate, and critique through self-reflection." (2024).
>
> [13] Zeng, W., Huang, Y., Liu, Q., Liu, W., He, K., Ma, Z., & Junxian He. (2025). SimpleRL-Zoo: Investigating and Taming Zero Reinforcement Learning for Open Base Models in the Wild. *Proceedings of COLM 2025 / OpenReview*.

---

### Official Review · Reviewer_LVGE · 2025-10-31

**Soundness:** 1
**Presentation:** 1
**Contribution:** 1
**Rating:** 2
**Confidence:** 4

**Summary:**

This paper proposes UR^2, a reinforcement learning (RL) framework that unifies retrieval-augmented generation (RAG) and reasoning. The authors point out two limitations in prior RAG-RL frameworks (e.g., *Search-R1*): (1) existing models fail to find an optimal trade-off between retrieval and reasoning, often overemphasizing one component; (2) retrieval introduces significant noise, which can degrade chain-of-thought reasoning quality.

To address these issues, the paper introduces two main components: (1) a hybrid retrieval corpus that combines domain-specific offline corpora with LLM-generated summaries, and (2) a difficulty-aware curriculum, where retrieval is invoked only for hard instances while easy ones rely on internal reasoning.

Through experiments on various benchmarks (MMLU-Pro, MedQA, MATH, and open-domain QA), the authors report that UR^2 improves performance compared to existing RAG-RL methods and achieves comparable results to GPT-4o-mini with 7B-scale models.

**Strengths:**

- The idea of difficulty-aware curriculum and task-mixing strategy is interesting and intuitively appealing.
- The paper presents comprehensive evaluations across multiple domains (math, medicine, open QA).

**Weaknesses:**

- Fairness and comparability of evaluation. The proposed method introduces an augmented knowledge base (LLM-summarized and task-specific corpora) as a key component. However, previous RAG-RL methods (e.g., Search-R1, R1-Searcher) are not restricted to specific corpora and could also utilize similar augmented data. Therefore, evaluating UR^2 solely on newly curated corpora while prior methods rely on static Wikipedia datasets makes the comparison unfair. If the augmentation of the corpus itself is a core contribution, the paper should explicitly analyze the effect of this augmentation (e.g., via retrieval corpus ablation), similar to prior work such as [1].
- Limited novelty and weak justification of the method. The overall training pipeline remains very close to the Search-R1 style frameworks, merely adding a two-stage optimization process. However, the paper does not convincingly explain how this design resolves the original issue mentioned in the introduction (the imbalance between retrieval and reasoning). The conceptual connection between the two-stage RL setup and this problem statement is unclear.
- Unclear source of performance gain. The improvement might stem from broader training dataset coverage or the use of enhanced corpora rather than from the proposed algorithm itself. Because the training data distribution and retrieval sources differ across baselines, it is hard to isolate whether gains come from the method or from data and corpus augmentation. A controlled comparison under identical conditions would strengthen the claim.
- Lack of clarity and reproducibility in RL setup. The term “Vanilla RL” baseline is ambiguous. It's unclear whether it corresponds to a GRPO-based single-stage model (as in Search-R1) or a simplified UR^2 variant. Moreover, the use of REINFORCE++ instead of PPO or GRPO (common in previous works) is not justified or compared. Without such clarification, it is difficult to determine whether improvements arise from the proposed method design or simply from a change in the RL algorithm.

[1] Lyu et al., Frustratingly Simple Retrieval Improves Challenging, Reasoning-Intensive Benchmarks

**Questions:**

- What exactly does the Vanilla RL baseline represent? Does it follow the same architecture as Search-R1 with different data, or is it a simplified UR^2 variant?
- Did you compare REINFORCE++ against GRPO or PPO under the same setup? If not, how can we be sure that the gains are not due to the RL algorithm itself?
- How much of the reported improvement remains if all methods are trained on the same corpus and datasets? In particular, does UR^2 still outperform Search-R1 when using the same Wikipedia-based retrieval corpus?
- How are the LLM-summarized corpora evaluated in terms of factual accuracy and hallucination?Any quantitative evidence that they indeed reduce retrieval noise would be valuable.

---

> ### Author Response · Authors · 2025-11-13
> **Rebuttal to Reviewer LVGE, part 1**
>
> We sincerely thank the reviewer for the thoughtful feedback. Below we address each concern:
>
>
>
> **Q1: What is "Vanilla RL"?**
>
>
>
> **To ensure a fully fair comparison**, as stated in Lines 288–289, Vanilla RL is a baseline following the same setup and datasets as UR², applying RAG-RL to open-domain QA and CoT-RL to math/MCQ tasks. Specifically: (1) identical or subset training data, (2) REINFORCE++ algorithm, (3) same hyperparameters (KL coef, batch size), and (4) standard domain-specific training methods (CoT-RL in reasoning tasks and RAG-RL in open-domain QA). In the retrieval scenario, it serves as **a single stage multi-step search–reasoning training baseline similar to Search-R1, using the same setup and subset training data as UR²**. Our method achieves performance far superior to Vanilla-RL across almost all benchmarks on all models, demonstrating the superiority of **our proposed difficulty-aware curriculum and LLM-summarized corpora**.
>
> Regarding data consistency: In main experiments (Tables 1-2), Vanilla RL uses one dataset per domain, following current practice ([7,8,9]). Beyond unifying paradigms, our approach jointly optimizes **reasoning**- and **retrieval**-oriented **data** to strengthen both tool-use and reasoning abilities, enabling mutual enhancement across tasks-a key distinction from Vanilla RL. The consistent advantages across settings confirm gains stem from our proposed method. As stated in Lines 68-70, UR² outperforms previous  SOTA methods by **5.8% (7B) and 19.0% (3B)** on average, with peak gains of 9.5% and 29.6%.
>
> If you want to see the impact of the corpus summarization module and the multi-step search–reasoning paradigm: Table 8 presents Vanilla RL MCQ with **identical training data** as UR² (Lines 754-755), yet UR² shows significant improvements, highliting the superiority of summarization module and multi-step search–reasoning paradigm in such reasoning scenarios.
>
>
>
> **Q2: REINFORCE++ vs GRPO/PPO comparison?**
>
>
>
> See Response to W4.
>
>
>
> **Q3: Would UR² outperform Search-R1 on identical data/corpus?**
>
>
>
> One key insight of our work is that **incorporating a summarization module during training helps the model learn to use the retrieval tool more effectively**. On one hand, the w/o LLM Summary variant in Table 4 highlights the summarization module’s importance for learning retrieval usage in reasoning scenarios; on the other hand, the comparison between UR² and the 4o-mini/Qw3-8B Summary variants demonstrates that a noise-free retrieval environment during training significantly enhances the model’s ability to utilize the retrieval tool.
>
> Therefore, our method **inherently includes summarization module and dataset diversity** (see Q1 response and the paragraph above). The component-wise improvements are clearly reflected in Table 5 and Table 9, as well as in the bottom-left part of Figure 1, where the performance increases from 33.8 (baseline) → 50.6 (+RL) → 57.5 (+LLM-summarized corpus) → 58.5 (ours). Requiring identical data/corpus as Search-R1 misses our core contribution. Notably, we use only \~1/3 of R1-Searcher's RAG data (Lines 996, 1013) yet achieve significant gains (60.4 vs 57.0 in Table 2). Moreover, our method’s substantial improvement over the Vanilla baseline in Table 2 further demonstrates its superiority.

---

> ### Author Response · Authors · 2025-11-13
> **Rebuttal to Reviewer LVGE, part 2**
>
> **Q4: How to assess LLM summary accuracy and noise reduction?**
>
> First, we clarify that all summarization is performed **online**, rather than using LLMs to pre-clean the corpus, as clearly illustrated in Fig. 1 and Section 3.1.1.
>
> We believe this aspect **has been sufficiently analyzed in prior and concurrent works**, and the effectiveness is **also reflected in our experimental results** (Tables 4, 10-12). As shown in Table 10, downstream performance consistently improves with stronger summarization models, indicating that LLM-generated summaries effectively enhance signal quality and reduce retrieval noise. Therefore, we did not repeat the same qualitative analysis in our experiments.
>
> Specifically, EDC²-RAG [3] provides a detailed examination of how model-based summarization mitigates retrieval noise and improves factual consistency. Concurrent studies such as WebSailor-V2 [1] and Beyond Ten Turns [2] also adopt external summarization modules instead of directly feeding noisy long contexts to LLMs, further confirming the effectiveness and necessity of this design.
>
> Additionally, we randomly sampled 10,000 entries from offline retrieval intermediate files and calculated compression ratios using the Qwen-2.5-7B-Instruct tokenizer. For offline retrieval: average tokens before→after: 1,291→128 (compression ratio: 7.87×). For online retrieval (3,000 samples): 11,346→761 (compression ratio: 14.91×), demonstrating the importance of the summarization module.
>
> Furthermore, our submitted **Supplementary Material** provides extensive evaluation results, through which the influence of LLM-summarized information on reasoning steps can be clearly observed, along with corresponding prompts in **Appendix D.3** and illustrative examples in **Appendix E**, demonstrating how summaries preserve factual information while filtering out irrelevant content.
>
> **If further clarification is desired, we are happy to include additional quantitative evaluations of factual accuracy and hallucination rates.**
>
>
>
> **W1: Fairness of evaluation & corpus enhancement**
>
>
>
> Lines 297-299 clarify all models use GPT-4.1 for summarization, including **RAG-RL baselines** (Search-R1, R1-Searcher). Lines 294-297 detail corpus/retriever settings. We ensure strict fairness and comparability across all evaluations. All methods share the **identical** evaluation setup, including the summarization model, corpus, retriever, and top-k configuration.&#x20;
>
> Table 11 compares three RAG-RL methods with ours across four **identical** retrieval configurations.Our method is slightly weaker than Search-R1 and R1-Searcher only under the *FULL corpus without summarization* setting, which is expected: when processing long texts, noise handling is delegated to the summarization model during training, leading to a slightly lower robustness of the policy model itself. Nevertheless, as noted in Lines 824–826, our approach achieves better performance with fewer retrievals than Search-R1, making it more efficient and deployable in real scenarios. Similar concurrent works (e.g., WebSailor-V2 [1] and Beyond Ten Turns [2]) also adopt external summarization modules rather than letting the LLM directly handle noisy long contexts, further validating our design choice.
>
> Corpus enhancement by **summarization module** is indeed a core contribution. Therefore, we provide extensive ablations in both the training and evaluation stages: Appendix B.3, Tables 3, 4, and 9. As stated in Lines 441-446, high-quality training corpus (GPT-4.1-mini Summarization) critically impacts tool-use ability—even with identical evaluation settings, methods differ significantly. To clarify, all summarization is performed **online**, rather than using LLMs to pre-clean the corpus, as clearly illustrated in Fig. 1 and Section 3.1.1, and we did **not** construct an overly **complex corpus** or apply sophisticated retrieval or preprocessing. Our corpus consists of only four components: Wikipedia Full (for reasoning tasks), Wikipedia Abstract (for open-domain QA), OnlineSearch, and Textbooks from MedQA [6] (for ablation in Table 12).

---

> ### Author Response · Authors · 2025-11-13
> **Rebuttal to Reviewer LVGE, part 3**
>
> **W2: Limited novelty & unclear motivation**
>
> We appreciate the reviewer’s comments and would like to clarify the motivation and contribution of our framework.
>
> While the overall training pipeline shares a similar multi-step search–reasoning framework with prior Search-R1–style methods, our key innovation lies in unifying retrieval and reasoning through a two-stage optimization framework that integrates **corpus design, training data processing, and curriculum learning**. Table 4 validates our claims on **retrieval-reasoning imbalance**: (1) W/o P\_fallback penalty: Allows unrestricted queries to summarizer (offloading reasoning), resulting in performance drop—confirming over-reliance on retrieval harms downstream tasks. (2) W/o LLM Summary: Validates noise issues (Lines 47-48) causing retrieval failure in reasoning scenarios.
>
> Our method addresses this issue in two key ways: (1) By retaining a portion of simple and medium-difficulty cases for direct reasoning during training, we preserve the model’s intrinsic **reasoning ability**, while the use of strict summarization prompts and the *P\_fallback* penalty prevents over-reliance on retrieval. (2) The summarization module effectively eliminates noise from retrieved passages and, through the stage-1 warm-up process, stimulates the model’s **retrieval capability**. Overall, our design directly addresses this concern, enabling the model to **perform retrieval and reasoning in a more intelligent and balanced manner** (see examples in Figure 4).
>
> We also emphasize that the **summarization module** is not a mere auxiliary component but an **essential mechanism for real-world applicability**. In dynamic online retrieval environments, documents are often lengthy and noisy. Our empirical analysis using the Qwen2.5-7B tokenizer shows a compression ratio of 14.9× (11,346→761 tokens), effectively reducing computational overhead (O(n²) complexity in attention) and preventing severe performance degradation. As detailed in Section 6.1, our method even outperforms offline retrieval in online scenarios, confirming its robustness in noisy settings. Moreover, concurrent systems such as WebSailor-V2 [1], Beyond Ten Turns [2], and Tongyi DeepResearch [5] adopt similar multi-step search–reasoning frameworks with **summarization** and RL-based incentives, achieving strong performance on DeepResearch-type benchmarks. **These trends further support the scalability and timeliness of our design**. Moreover, our work is the **first** to elevate the **summarization module to a central role in multi-step search–reasoning scenarios**, which itself provides an important insight for future system design.
>
> Furthermore, we would like to highlight our method’s strong performance on **reasoning tasks**, which represent a core contribution of this work. As stated in Lines 43–45 (”*Most methods focus narrowly on open-domain QA..."*), our approach substantially outperforms CoT-RL SOTA methods. For example, on medical reasoning tasks, our method outperforms previous CoT-RL SOTA methods, achieving 66.2 vs. 60.8 on Qwen2.5-7B and 58.3 vs. 49.7 on Qwen2.5-3B, demonstrating clear performance advantages.
>
> In summary, while our framework leverages established techniques (LLM summarization, curriculum learning), its **integration into a unified retrieval–reasoning pipeline** and validation across diverse reasoning tasks mark a substantial and practical advancement. The summarization module, in particular, is indispensable for enabling LLMs to handle complex, noisy, and dynamic environments—bridging the gap between controlled training and real-world deployment.
>
> **W3: Unclear performance gain source**
>
> See Q1 response and Q3 response. As stated in Lines 263-265, our data comes from SimpleRL-Zoo and R1-Searcher. Using \~1/4 of full data achieves comparable results, proving our curriculum mechanism's efficiency. Table 13 lists data sources for all methods. Experiments and Ablations across settings consistently show significant advantages over baselines. We have conducted the corresponding experiments and ablations for almost all pairwise model comparisons that can be found in Table 13.
>
> If desired, we are willing to provide a dedicated table linking each ablation component to its corresponding experiment in future versions.

---

> ### Author Response · Authors · 2025-11-13
> **Rebuttal to Reviewer LVGE, part 4**
>
> **W4: RL setup clarity & reproducibility**
>
> See Q1 and W3 responses.
>
> About reproducibility concerns: **Sections 4.1** and **Appendix C** provide detailed hyperparameter specifications and training setups, while **Section D** lists all prompts used during training. In addition, our submitted **Supplementary Material** includes comprehensive evaluation results and training scripts. We believe **these are sufficient to reproduce our work**. In fact, we have successfully deployed our training framework on two 8×A100 servers with different CUDA versions and one 8×H20 server, which strongly demonstrates its reproducibility. Furthermore, we will release all code, models, and data upon acceptance.
>
> Regarding algorithmic differences between GRPO and REINFORCE++, we emphasize three points:
>
> 1. **Minimal performance gap:** As shown in R1-Searcher’s Table 3 [4], GRPO and REINFORCE++ yield nearly identical results, indicating limited sensitivity to the RL algorithm choice.
>
> 2. **Stable optimization:** As described in Section 3.2, in the absence of KL or clip constraints, our method only differs from GRPO by incorporating batch normalization (Eq. 3), which improves stability and enables training.
>
> 3. **Fair comparison:** As discussed in Lines 188–192 and 295–297, we adopt R1-Searcher’s REINFORCE++ implementation and ensure comprehensive, fair comparisons under identical settings.
>
> Overall, we believe the **RL algorithm choice has minimal impact on the reported results**.
>
>
>
> Your concerns about component contributions and fairness were comprehensively addressed in our experimental design. Due to training and evaluation diversity, some experiments and cases are in the appendix, causing potential clarity issues. **We hope this response clarifies the importance and value of our contributions**. Thank you for the valuable feedback that helps improve our work!
>
>
>
> **References:**
>
> [1] Li, Kuan, et al. "Websailor-v2: Bridging the chasm to proprietary agents via synthetic data and scalable reinforcement learning." *arXiv preprint arXiv:2509.13305* (2025).
>
> [2] Gao, Jiaxuan, et al. "Beyond ten turns: Unlocking long-horizon agentic search with large-scale asynchronous rl." *arXiv preprint arXiv:2508.07976* (2025).
>
> [3] Li, Weitao, et al. “Efficient Dynamic Clustering-Based Document Compression for Retrieval-Augmented-Generation.” Findings of the Association for Computational Linguistics: EMNLP 2025, pp. 9833–9849. 2025. DOI: 10.18653/v1/2025.findings-emnlp.522.
>
> [4] Song, Huatong, et al. "R1-searcher: Incentivizing the search capability in llms via reinforcement learning." *arXiv preprint arXiv:2503.05592* (2025).
>
> [5] Team, Tongyi DeepResearch, et al. "Tongyi DeepResearch Technical Report." arXiv preprint arXiv:2510.24701 (2025).
>
> [6] Jin, Di, et al. "What disease does this patient have? a large-scale open domain question answering dataset from medical exams." *Applied Sciences* 11.14 (2021): 6421.
>
> [7] Goldie, Anna, et al. "Synthetic data generation & multi-step rl for reasoning & tool use." arXiv preprint arXiv:2504.04736 (2025).
>
> [8] Xi, Zhiheng, et al. "Agentgym-rl: Training llm agents for long-horizon decision making through multi-turn reinforcement learning." arXiv preprint arXiv:2509.08755 (2025).
>
> [9] Zhang, Kai, et al. "Agent Learning via Early Experience." arXiv preprint arXiv:2510.08558 (2025).

---

> ### Comment · Reviewer_LVGE · 2025-11-24
>
> Thank you for the detailed rebuttal and clarification. I now clearly understand that UR2 utilizes an on-the-fly summarization module rather than a pre-refined static corpus, and that the gains are not solely due to the RL algorithm choice.
>
> However, I maintain my **Rating at 2** (with an increased **Soundness** score) due to the following remaining concerns:
>
> 1. **The phrase "UR^2 employs a LLM-summarized retrieval corpus" (`L142-143`) is highly misleading**, causing 3 out of ~4~5 reviewers to mistake it for a static offline corpus. The paper must explicitly rephrase this (e.g., "employing an on-the-fly summarization module") and revise Figure 1 to clearly visualize the runtime flow.
>
> 2. **Using GPT-4.1 as a summarizer (`L296-298`) for 3B/7B policy models yields an unfair comparison against baselines.** This setup masks the significant computational cost and reliance on a proprietary model. While I acknowledge authors' response to Reviewer 8c2q regarding the potential use of distilled models, the default results in the main tables for a 3B/7B-scale should utilize a summarizer of comparable scale (e.g., 7B) to ensure a fair comparison. At the very least, all tables must explicitly state the use of GPT-4.1 as a summarizer to prevent misleading readers about the standalone capabilities of the 7B model.
>
> 3. Recent work [1] proposes a similar "Search and Refine" concept but integrates the refinement process into the policy model itself via RL, without relying on external summarizers. This approach appears more elegant and efficient. **Consequently, the contribution of UR2, which depends on an external module, seems limited in comparison.** While I acknowledge the proposed difficulty-aware curriculum and data selection strategies, I find these components to be relatively standard techniques (e.g., difficulty-aware curriculum learning in [2], data selection strategies in [3]) that do not sufficiently distinguish this work or outweigh the methodological limitations mentioned above.
>
> ### References
>
> [1] Shi et al., Search and Refine During Think: Facilitating Knowledge Refinement for Improved Retrieval-Augmented Reasoning, NeurIPS 2025
>
> [2] Parashar et al, Curriculum Reinforcement Learning from Easy to Hard Tasks Improves LLM Reasoning
>
> [3] Ye et al., LIMO: Less is More for Reasoning, COLM 2025

---

> ### Author Response · Authors · 2025-11-24
>
> Thanks for your response. But we have some concerns regarding your reply.
>
> For your first concern about the description of our implementation of summarization, we acknowledge this point and will refine it in the future version of the paper. However, we believe there is a **factual error** in your comment: this submission received **5** reviewers in the first review round (not **4**).
>
> Regarding the fairness of comparisons, we believe we have clearly stated in our prior response that **all RAG baselines also use GPT-4.1 as the summarizer, including RAG-RL baselines such as SEARCH-R1 YOU MENTIONED**.  As for your concern about clarifying this implementation detail in our paper, we believe that putting it in the main text (lines 296–298) is appropriate and standard practice.
>
> Concerning your comments on novelty, we want to clarify that we deliberately keep the noisy deduction part as an external module to allow the policy model to focus on reasoning, proposing retrieval queries and utilizing retrieved documents. These two approaches (the one mentioned in “Search and Refine” and ours) each have their own advantages. We have detailed these advantages in our previous response, such as reduced computational complexity and improved robustness in real-world online retrieval environments. We also acknowledge that some techniques we use may appear in prior works, but bringing them together in a **new area** is also valuable. We believe we have clearly explained our motivations: 1) The RAG-RL paradigm is limited in open-domain QA scenarios. 2) There is imbalanced retrieval and reasoning in works like R1-Searcher or Search-R1. 3) Retrieved documents often contain noise and redundancy. And our concise framework effectively addresses these issues with acceptable computational cost. Regarding the curriculum design, we want to emphasize that our method introduces an innovative design that **assigns different prompt strategies (CoT vs. RAG) to data of different difficulty levels and incorporates this distinction into training**. Experimental results show that this approach has a notably positive impact on addressing the problem of imbalanced retrieval and reasoning. We also observe similar ideas emerging in some concurrent commercial systems. We believe all results in our paper firmly support our viewpoint.
>
> Thank you again for your response. Your suggestions will help us improve our work.

---

> > ### Comment · Reviewer_LVGE · 2025-11-28
> >
> > Thank you for the detailed follow-up response. Thank you also for pointing out the factual mistake in my previous comment. I corrected it to "3 out of 5 reviewers" immediately after reading your response.
> >
> > I now understand that **all RAG baselines reported in Table 1 and 2 also use an LLM summarizer.** If my understanding is correct, previous related concerns regarding the evaluation fairness are resolved.
> >
> > With this clarified, I have a follow-up question related to this point. I noticed the ablation in L442–444 indicating that the *W/o LLM Summary variant fails and degenerates into pure CoT*, which you interpret as evidence that summarization is necessary for handling retrieval noise in RAG-RL settings. I would appreciate it if you could elaborate on this point in more detail. Specifically, **does the model trained with RL fail to use search at all when retrieved documents are not summarized?** If so, is the failure mainly due to the length of the retrieved documents or the quality of the retrieved documents? If it is due to length, does truncation not help? If it is due to quality, could you clarify why the retrieved passages are noisy despite originating from the same Wikipedia knowledge base?
> >
> > Regarding clarity, I appreciate the authors acknowledging this point and stating that it will be refined in a future version of the paper. I would like to note that authors can upload a revised paper during the rebuttal phase. Uploading the updated version would help reviewers more accurately assess how the rebuttal is incorporated into the paper.

---

> > > ### Author Response · Authors · 2025-11-28
> > >
> > > Thank you for your timely follow-up.
> > >
> > > ---
> > >
> > > Regarding the **W/o LLM Summary variant**, your understanding is correct. The core issue is that noise in retrieved documents prevents retrieval from providing consistent positive feedback during RL training. A simple example comes from the medical domain: under a vanilla RAG-RL setup similar to Search-R1, the policy may retrieve in some rollouts and skip retrieval in others. Due to noisy documents, the rollouts without retrieval can receive higher rewards, leading the policy to gradually suppress the retrieval action. **In other words, the model fails to obtain useful information from retrieval, so it eventually abandons this behavior during training**.
> > >
> > > We also ran an additional experiment on Qwen2.5-14B-Instruct using only the medical training data without any summarizer, with the top-k set to 2 (the maximum the model can reliably handle without overwhelming noise). In this case, the model was still able to trigger retrieval to some extent, achieving 72.4 (RAG) versus 69.8 (CoT). A stronger model thus partially compensates for noise. However, as shown in Table 5, even under the same evaluation and retrieval conditions, UR² substantially outperforms variants using GPT-4omini or Qwen3-8B. **This supports an important insight: a clean and stable retrieval environment enables the policy to learn to use the retrieval tool more effectively**. We also expect that the 14B model would benefit further if training were conducted with summarization, since a summarizer allows increasing top-k and exposing the model to more useful evidence.
> > >
> > > Table 10 also shows that UR² without summary at evaluation still surpasses Vanilla-RL by a large margin. It is an interesting phenomenon that the policy struggles to use retrieval during training without summarization but becomes more tolerant to noise after learning. We interpret this **as a typical case of policy hacking**, where the model learns to exploit an easier and more familiar strategy (CoT) during training, leading it to a local optimum even though retrieval is in fact beneficial.
> > >
> > > ---
> > >
> > > Regarding manuscript updates, several reviewers have not yet responded, and we currently **reference** line numbers and tables that **may shift after revisions**. We are also preparing to replace the main results by re-running the evaluation with a distilled Qwen2.5-7B summarizer. Once all first-round comments are received and the updated results are ready, we will revise the manuscript accordingly.
> > >
> > > ---
> > >
> > > Thank you again for your constructive suggestions. They have been very helpful, and we appreciate the opportunity to clarify these points.

---

### Official Review · Reviewer_8c2q · 2025-11-01

**Soundness:** 3
**Presentation:** 3
**Contribution:** 2
**Rating:** 6
**Confidence:** 2

**Summary:**

This paper introduces $UR^{2}$, a general framework that unifies Retrieval-Augmented Generation (RAG) and reasoning through reinforcement learning to overcome the limited scope of prior RAG-RL methods. The framework's key innovations are a difficulty-aware curriculum that selectively invokes retrieval only for challenging problems and a hybrid knowledge access strategy that combines domain-specific offline corpora with LLM-generated summaries. Experiments across diverse tasks, including open-domain QA, MMLU-Pro, medical, and mathematical reasoning, show that $UR^{2}$ significantly outperforms existing RAG and RL methods on several benchmarks.

**Strengths:**

- The paper addresses a well-defined and significant problem: the suboptimal trade-off between retrieval and reasoning in current models . The core idea of a difficulty-aware curriculum is an intelligent and novel solution. It directly trains the model to learn when to trust its parametric knowledge versus when to seek external knowledge, a critical capability for building more efficient and robust agents.

- The two-stage optimization process is a key strength . Stage 1 provides a clever solution to the "cold start" problem often seen when training RL policies for tool use, by first rewarding the model for simply learning the tool-calling format. Decoupling this from Stage 2 (answer correctness) likely leads to a more stable training signal and better credit assignment.

- The experimental validation is exceptionally thorough and is a major strength of this work.

**Weaknesses:**

- As the authors acknowledge, the experiments are limited to 8B-parameter models. If scaling to larger models is too costly, I wonder whether smaller models can also benefit from the proposed method. Since using retrievers or tools with small models can help mitigate limited memorization capacity of the smaller models [1,2], it would be important to include experiments on smaller models.

- The curriculum design, while effective, introduces a significant one-time computational cost. To categorize questions by difficulty, the authors must "perform 20 rollouts using Qwen-2.5-7B-Instruct and compute the average performance score" for each training sample. This is a substantial preprocessing step that adds overhead before the main RL training can begin.

- The framework's robustness relies heavily on the "LLM-summarized retrieval corpus" to handle noise, as shown by the w/o LLM Summary ablation. This introduces a dependency on another powerful LLM (e.g., GPT-4.1) during the data preparation or inference pipeline, which adds complexity, latency, and cost that are not fully accounted for in the primary model's performance.


[1] Kang, Minki, et al. "Knowledge-augmented reasoning distillation for small language models in knowledge-intensive tasks." Advances in Neural Information Processing Systems 36 (2023): 48573-48602.


[2] Kang, Minki, Jongwon Jeong, and Jaewoong Cho. "T1: Tool-integrated self-verification for test-time compute scaling in small language models." arXiv preprint arXiv:2504.04718 (2025).

**Questions:**

Please see the weaknesses above.

---

> ### Author Response · Authors · 2025-11-13
> **Rebuttal to Reviewer 8c2q, Part 1**
>
> We sincerely appreciate your recognition of our work’s value, experimental rigor, and your constructive questions, which are crucial for further improving our work. We address each concern below.
>
> **W1: Model Scale and Generalization**
>
> We appreciate your concern about scalability. We have conducted extensive experiments on smaller models, including **Qwen2.5-3B**, with detailed baselines and ablations (Tables 6–8) and qualitative curves (Fig. 3). As stated in lines 314–317: “*Across model scales, UR² shows consistent advantages: on Qwen-2.5-3B, it achieves even larger gains (+9.1% on MMLU-Pro and +8.6% on Medicine over Vanilla RL), demonstrating that UR² provides greater benefits for models with limited knowledge but strong reasoning capabilities.*”These results confirm that the proposed method benefits both small and larger (7B or 8B) models.
>
> If you request experiments on smaller models, it would be challenging, as our method does **not** rely on any **expert demonstrations** or **model distillation** but instead focuses on stimulating the model’s own capability. This makes it difficult for smaller models to perform well. We also view this as **one of the highlights of our work**. However, if you would like to see more detailed experiments, we would be glad to provide corresponding results using an SFT-based warm start.
>
> **W2: Curriculum Cost and Practicality**
>
> Since the rollout process is entirely offline, the cost is controllable compared with the RL stage. The difficulty labels can be generated more efficiently using (1) fewer rollouts, (2) smaller models, or (3) the existing difficulty annotations provided in our source datasets, i.e., SimpleRL-Zoo [4] (math) and R1-Searcher [1] (open-domain QA). We unified labeling via our own rollouts for methodological consistency. Comparable performance on Qwen2.5-3B-Instruct and Llama-3.1-8B-Instruct indicates that different rollout models or label sources do not significantly affect results.
> Furthermore, Table 5 validates the effectiveness of our curriculum design. As stated in Lines 464–465:  “*Despite using significantly less data, filtered datasets achieve comparable or superior performance, particularly on out-of-domain tasks.*”
>
> Moreover, using rollout models for difficulty annotation is a **standard practice** [1-3].

---

> ### Author Response · Authors · 2025-11-13
> **Rebuttal to Reviewer 8c2q, Part 2**
>
> **W3: Robustness and LLM-based Summarization**
>
> Our framework’s robustness partly depends on LLM-based summarization for noise handling. Regarding cost, as described in lines 1193–1197 (consumption detailed in lines 1191–1193): “*Since the training is a one-time expense, we consider the overall training-related consumption acceptable. Furthermore, experiments reported in Section B.3 and Table 4 show that substantial performance gains can be achieved without relying on closed-source models, suggesting that open-source models or less expensive APIs provide a viable alternative for achieving comparable improvements.*”
>
> Concurrent studies such as WebSailor-V2 [5] and Beyond Ten Turns [6] adopt similar summarization modules in long-context retrieval scenarios, confirming its necessity. We thus view it as a **reasonable trade-off** — a modest cost for stronger robustness and significant downstream gains.
>
> Furthermore, it is a necessary part under online retrieval environment: in online retrieval environments, documents are often excessively long. We calculated token counts using the **Qwen-2.5-7B-Instruct** tokenizer and randomly sampled 3,000 entries from saved intermediate results:
>
> > Compression before → after: **11,346 → 761** (compression ratio: **14.91×**)
>
> Directly concatenating them into prompts causes rapid context length expansion, leading to computational waste and significant performance degradation. Given the **O(n²)** complexity of the attention mechanism, this creates enormous difficulties for model inference and training. Using a summarization module or a new context window to process noise can significantly reduce this overhead **by nearly an order of magnitude**.
>
> Thank you again for your insightful feedback and recognition of our work’s value. If there are any additional analyses you deem necessary, we will gladly provide them. We have great respect for your responsibility, professionalism, and constructive questions demonstrated during the review process.
>
> **References:**
>
> [1] Song, Huatong, et al. "R1-searcher: Incentivizing the search capability in llms via reinforcement learning." arXiv preprint arXiv:2503.05592 (2025).
>
> [2] Huang, Jerry, et al. "Rag-rl: Advancing retrieval-augmented generation via rl and curriculum learning." arXiv preprint arXiv:2503.12759 (2025).
>
> [3] Sun, Yifan, et al. "Improving Data Efficiency for LLM Reinforcement Fine-tuning Through Difficulty-targeted Online Data Selection and Rollout Replay." arXiv preprint arXiv:2506.05316 (2025).
>
> [4] Zeng, W., Huang, Y., Liu, Q., Liu, W., He, K., Ma, Z., & Junxian He. (2025). SimpleRL-Zoo: Investigating and Taming Zero Reinforcement Learning for Open Base Models in the Wild. Proceedings of COLM 2025 / OpenReview.
>
> [5] Li, Kuan, et al. "Websailor-v2: Bridging the chasm to proprietary agents via synthetic data and scalable reinforcement learning." arXiv preprint arXiv:2509.13305 (2025).
>
> [6] Gao, Jiaxuan, et al. "Beyond ten turns: Unlocking long-horizon agentic search with large-scale asynchronous rl." arXiv preprint arXiv:2508.07976 (2025).

---

> ### Author Response · Authors · 2025-11-21
> **Rebuttal Update: Distilled 7B Summarizer Experiment**
>
> To further verify that our method does not rely on a strong summarization model, we reran the entire pipeline using a **distilled Qwen2.5-7B-Instruct summarizer**, trained on 90k×2-epoch intermediate summaries generated by GPT-4.1-mini (20k per MMLU-Pro subject, 30k for Medicine). The SFT process required 156 hours of H20 GPU time (8×19.5h). The RL configuration and all evaluation settings followed the original UR² setup.
>
> The results are below:
>
> | Method                    | Hist | Phil | Econ | Law  | Avg  | MedQA | M-Med | Avg  |
> |-----------------|------|------|------|------|------|-------|------|------|
> | **CoT**                   | 42.3 | 45.7 | 63.4 | 26.6 | 44.5 | 57.2  | 52.0  | 54.6 |
> | **Standard RAG**          | 44.6 | 41.1 | 57.8 | 26.0 | 42.4 | 54.2  | 53.2  | 53.7 |
> | **General Reasoner**      | 47.9 | 44.2 | 65.9 | 30.4 | 47.1 | 58.4  | 54.4  | 56.4 |
> | **Open-Reasoner-Zero**    | 50.0 | 46.6 | 67.5 | 34.2 | 49.6 | 61.6  | 58.8  | 60.2 |
> | **SimpleRL-Zoo**          | 35.7 | 36.9 | 55.2 | 25.4 | 33.8 | 57.2  | 51.0  | 54.1 |
> | **Vanilla RL**            | 52.2 | 43.5 | 64.0 | 33.8 | 48.4 | 64.2  | 57.4  | 60.8 |
> | **UR² (Original)**        | 53.2 | 53.0 | 72.2 | 35.0 | **53.3** | 69.6  | 62.8  | **66.2** |
> | **UR² (Distilled Qwen2.5-7B)**  | 49.6 | 52.3 | 68.2 | 31.4 | ***50.4*** | 67.2  | 61.8  | ***64.5*** |
> | **UR² (4omini)**        | 49.3 | 48.8 | 67.4 | 32.4 | 49.5 | 65.0  | 59.2  | 62.1 |
> | **UR² (Qw3-8B)**        | 49.1 | 49.9 | 67.8 | 30.6 | 49.4 | 64.8  | 58.2  | 61.5 |
>
> Despite using a significantly weaker summarizer, UR² still **outperforms all baselines across all domains**, including General Reasoner, Open-Reasoner-Zero, SimpleRL-Zoo, and Vanilla RL.
>
> These findings support two key conclusions:
>
> 1. **UR² remains effective even with a much lighter summarization model.**
>    Using a distilled 7B summarizer preserves strong performance while substantially reducing computational cost, demonstrating that the method is practical, efficient, and easy to deploy in real-world settings.
>
> 2. **The gains stem from UR²'s framework design rather than summarizer capacity.**
>    The method benefits primarily from a **clean, low-noise retrieval environment** produced by summarization, which in turn helps the policy model exhibit more reliable retrieval behavior. This aligns with our claim that summarization acts as environment shaping, not as a teacher model.
>
> We hope this additional experiment directly addresses your concern and clarifies that **UR² remains effective, robust, and practically deployable even with a lightweight summarizer**.

---

### Author Response · Authors · 2025-11-14
**Formal Clarification — Online, On-the-Fly Summarization (Not Offline Corpus Preprocessing)**

This clarification is addressed **specifically to Reviewers LVGE, 46ub, and nM4H**.

We emphasize the following core fact of our system design: **UR² uses *online, on-the-fly* LLM summarization triggered at retrieval time. We do *not* pre-clean, pre-summarize, or pre-construct any offline summarized corpus.**

All summaries are produced dynamically **after** documents are retrieved, during training or inference, as part of the multi-step reasoning–retrieval loop. Summaries are *not* stored, curated, or used as a static data resource.

We respectfully request that all evaluations, concerns, and judgments regarding fairness, corpus augmentation, robustness, or corpus construction be reconsidered **under this correct premise**.

---

**Explicit Locations in the Paper (Cited Exactly)：**

(1) Fig. 1 and Section 3.1.1：These explicitly show the retrieval → online summarization → reasoning pipeline *as clearly illustrated in Fig. 1 and Section 3.1.1.*

(2) Online Retrieval Implementation (lines 1052–1091): Online HTML → Markdown → **Content Summarization** performed by GPT-4.1, proving the summarizer is an online runtime module, not an offline corpus generator.

(3) Training-time Summarization Prompt (Appendix D.3): Appendix includes the exact summarization prompts used **during training**, confirming that summaries are created per retrieval call.

---

**Kind Request for Re-evaluation:**

Given the explicit evidence above, we kindly request that reviewers consider these points and reconsider the associated ratings:

* *LLM-summarized corpora* being treated as a **static** offline corpus
* fairness and comparability
* robustness to dynamic or real-world retrieval
* misinterpretation of summaries as corpus augmentation rather than a runtime tool

These concerns rely on a premise that is **inconsistent** with the implementation clearly shown in the cited sections.

---

Because the work spans multiple datasets, retrieval modes, and multi-stage training workflows, providing every detail repeatedly in the main text would significantly disrupt readability. Some details were therefore placed in figures and appendices, which may have led to partial misunderstanding. We will ensure that future versions use clearer wording to avoid such ambiguity. The cited sections above fully specify the online nature of summarization.

---

### Comment · Area_Chair_vseU · 2025-11-24
**Reviewer & Author Discussion**

Hi Reviewers,

Please kindly and actively participate in the review-author discussion if you haven't already, raise your further concerns so that the authors can explain more, and make your final decisions.

Best,
AC

---

### Note · Authors · 2025-12-10

I have read and agree with the venue's withdrawal policy on behalf of myself and my co-authors.